# Detection of Falls and Frailty in Older Adults with Oldfry: Associated Risk Factors

**DOI:** 10.3390/s25102964

**Published:** 2025-05-08

**Authors:** Eva Martí-Marco, Enrique J. Vera-Remartínez, Aurora Esteve-Clavero, Irene Carmona-Fortuño, Martín Flores-Saldaña, Jorge Vila-Pascual, Malena Barba-Muñoz, María Pilar Molés-Julio

**Affiliations:** 1Faculty of Health Sciences, Department of Nursing, Universitat Jaume I, 12006 Castellón, Spain; al389241@uji.es (E.M.-M.); estevea@uji.es (A.E.-C.); icarmona@uji.es (I.C.-F.); saldana@uji.es (M.F.-S.); al452755@uji.es (M.B.-M.); 2Provincial Hospital of Castellón, 12003 Castellón, Spain; 3Penitentiary Centre “Castellón I”, 12006 Castellón, Spain; enriquevera@gmx.es; 4General Hospital of Castellón, 12004 Castellón, Spain; 5Artana Health Centre, 12527 Castellón, Spain; 6Nursing Home of Caritas in Burriana, 12530 Castellón, Spain; jorgevilapascual@gmail.com

**Keywords:** older adult, frailty, falls, factors, tool

## Abstract

**Highlights:**

**What are the main findings?**

**What is the implication of the main finding?**

**Abstract:**

Objective: To describe the characteristics and outcomes of using the Oldfry technology application in older adults, analyzing changes in frailty and fall risk after its implementation. Design and Methods: Observational, analytical, prospective, cross-sectional, and multicenter study conducted in residential centers in Plana Baja (Castellón, Spain). A total of 156 older adults over 65 years old participated, selected based on specific criteria and voluntary consent. Sociodemographic, anthropometric, and clinical variables were collected, including fall history, sensory problems, medication use, and standardized cognitive, nutritional, and functional assessment scales. The study was approved by the Ethics Committee of Universitat Jaume I. Results: The sample included 156 individuals (median age: 84 years). Women showed greater functional dependence (Barthel scale) and cognitive impairment (Pfeiffer scale). The Oldfry device detected frailty with statistically significant differences. A direct relationship was found between greater functional dependence and higher fall risk, as well as between higher comorbidity and increased fall risk. An adequate nutritional status was associated with a lower fall risk. Conclusion: The use of Oldfry is crucial for assessing frailty and fall risk in older adults. Factors such as functionality, comorbidities, and nutritional status directly influence fall prevention, highlighting the importance of technological tools in monitoring these risks.

## 1. Introduction

Frailty in older adults is a geriatric syndrome characterized by a decline in the ability to cope with physical and emotional stressors, significantly increasing the risk of falls, a public health problem with profound implications [1,2]. Falls, in addition to being a common cause of severe injuries, can lead to increased dependency, institutionalization in residential care facilities, and, in the worst case, premature death [3]. In Spain, it is estimated that 49.4% of older adults report a fear of falling, a concern that is associated with gait disorders, balance deficits, and previous falls [4]. This fear creates a vicious cycle, as it can limit mobility and increase the risk of further falls.

Identifying fall risk in older adults is crucial for preventing these events and their adverse consequences. Factors such as cognitive impairment, depression, reduced functionality, and nutrition play a key role in frailty and fall risk [5]. For example, depressive disorder increases the likelihood of falls by 1.62 times, regardless of other risk factors [6]. Similarly, frailty, measured through functional and cognitive components, negatively impacts the ability to perform basic activities of daily living, as suggested by the Barthel scale [7]. Individuals with greater functional dependence are more likely to experience falls [8].

In the nutritional domain, diet also plays a critical role in preventing frailty. Promoting the adoption of healthy diets, such as the Mediterranean diet, and reducing the consumption of ultra-processed foods has been associated with a lower prevalence of frailty [9]. Assessments like the MNA (Mini Nutritional Assessment) scale emphasize the need for a personalized nutritional approach to prevent malnutrition, particularly in pre-frail and frail groups, which represent 90.7% of this population [10]. Managing polypharmacy and comorbidities is also essential, as the combination of multiple medications and chronic diseases significantly increases the risk of sarcopenia, falls, and loss of autonomy [11,12].

A promising approach to addressing these challenges is the use of technologies that enable early detection of frailty and monitoring of its associated factors. The development of technological applications that assess frailty through various parameters, such as muscle strength, balance, mobility, and cognitive status, allows healthcare professionals to identify specific areas of weakness. These applications not only provide real-time diagnoses but also enable direct intervention through personalized rehabilitation programs. These programs are designed to strengthen the weaker aspects of each individual, such as muscle strength or balance, through specific exercises and fall prevention strategies. This proactive approach can significantly improve the quality of life for older adults by reducing the risk of falls and promoting healthier, more autonomous aging [13,14,15].

Given the impact of these factors, healthcare professionals must implement preventive strategies that address not only the physical risk of falls but also the nutritional, functional, and cognitive components, supported by technology, to improve care and reduce the consequences of frailty in the daily lives of older adults.

The Oldfry device is a tool that uses inertial sensors and timed tests to assess parameters such as balance, gait, strength, and reaction times, among other aspects. It is a more comprehensive technology than the traditional tests used to manually evaluate fall risk and frailty. Based on the information provided by this device, personalized prevention plans can be developed, anticipating potential risks and improving the quality of life for older adults.

The aim of this study is to describe the characteristics and outcomes associated with the use of the Oldfry technological application in older adults, focusing on the observed changes in frailty and fall risk levels after its implementation, in order to understand its potential as a tool for the evaluation and monitoring of these factors.

## 2. Materials and Methods

Study Design and Setting:

This is an observational, analytical, prospective, cross-sectional, multicenter, and non-randomized study aimed at evaluating the impact of the Oldfry technological application on reducing frailty and fall risk in older adults.

The study was conducted in residential care centers for older adults located in the regions of Plana Baja (Burriana, Nules, Moncofa, Onda, and Vila-real) and Plana Alta (Almazora and Castellón) in the province of Castellón, Valencian Community, Spain.

The population of individuals over 65 years old in Castellón is approximately 379,426 people. The selected centers offer a total of 2489 authorized residential places, according to the registry in November 2023, of which 1960 places (78.75%) are part of the Valencian Public System of Social Services.

The sample was obtained from 809 available places across the 9 centers: 4 social initiative centers with concerted action, 4 commercial initiative centers with contract-funded places, and 1 local entity center with contract-program funded places. The initial data collection was conducted between February and April 2024.

Participants and Sample:

The sample consisted of 156 participants, all over 65 years of age, selected through specific criteria and voluntary acceptance to participate in the study. The selection process involved initial phone contacts with the centers to assess their interest and availability for participation. Subsequently, in-person meetings were held to explain the details of the project, inclusion and exclusion criteria, as well as the assessment methodology and subsequent individualized physical activity intervention.

The study will be carried out in three phases:Validation of the tool: completed and pending publication (in press).Data collection and intervention to indicate specific exercises for 6 months (current phase or preliminary analysis).Subsequent analysis to evaluate the intervention performed (pre–post).

Inclusion Criteria:
Reside in the center and be ≥65 years old.No severe cognitive impairment that hinders measurement.No total dependency that hinders measurement.

Exclusion Criteria:
Desire not to participate or lack of informed consent from the participant or their legal guardian.

Sample Size:

The same sample from the Oldfry tool validation study was used: a total of 156 individuals.

Variables and Instruments Used:

Preliminary data from the validation of the Oldfry device were obtained from a validation study currently in press. To assess the validity of the device, a diagnostic analysis was performed, yielding a specificity of 72.7% for the identification of fall risk and a sensitivity of 91.9% for the detection of frailty. In terms of reliability, strong linear correlation coefficients were observed, with values of 0.7 and 0.9 when comparing the device’s output to conventional manual tests. Additionally, high intraclass correlation coefficients (ICCs) were found, all exceeding 0.8, indicating a high degree of measurement consistency.

Sociodemographic variables (age, sex, residence center) and anthropometric variables (weight, height, and Body Mass Index (BMI)) were collected. Additionally, variables such as fall history over the last year, vision and/or hearing problems, number of medications, and type of consumption were included. The assessment scales used were the following:Yesavage Depression Scale: Assesses depression [16].Barthel Scale: Evaluates functional dependency level [17].Pfeiffer Scale: Assesses cognitive level [18].Mini Nutritional Assessment (MNA): Evaluates the nutritional status of older adults [19].Timed Up & Go (TUG): Measures fall risk [20].Short Physical Performance Battery (SPPB): Evaluates frailty risk [21].

Data Collection:

Data were collected exclusively by the principal investigator, ensuring participant anonymity through assigned codes. Only the center’s reference professional knows the identity linked to each code.

Data Development and Analysis Procedure:

Information was collected from each patient’s medical records and through results provided by the Oldfry technological device.

Initially, a descriptive analysis of the main variables was carried out. For quantitative variables, the type of distribution was assessed using the Kolmogorov–Smirnov normality test. Those that followed a normal distribution were expressed through the mean and the 95% confidence interval. Non-normally distributed variables were expressed through the median and interquartile range, also considering minimum and maximum values.

Subsequently, a bivariate analysis was conducted using classical statistical techniques, relating the means of quantitative variables using Student’s T-test for independent samples with a normal distribution when comparing 2 different means, and the Mann–Whitney U-test when the variables did not follow a normal distribution.

Qualitative or categorical variables were compared using a non-parametric test such as the Pearson Chi-square test, comparing each category with the others. Fisher’s test was used for sizes smaller than 5 in some cells. 

A binary logistic regression model analysis was performed, considering as the dependent variable having experienced a fall during the last year and as independent variables those that showed statistical differences in the classical comparison.

A significance alpha level of less than 0.05 was adopted in all cases. IBM SPSS Statistics v.25 was used for statistical analysis of the variables.

Ethical Considerations:

This study was approved by the Ethics Committee of Universitat Jaume I (UJI) (CEISH/44/2024), in accordance with the Organic Law 3/2018 on the Protection of Personal Data and the Guarantee of Digital Rights, the Law 41/2002 of 14 November on Patient Autonomy, and Law 14/2007 of July 3 on Biomedical Research.

All participating centers signed consent as external entities, accepting participation in the study and understanding the aspects to be developed. Each participant gave written informed consent to participate in the research.

## 3. Results

The sample consisted of 156 individuals of both sexes, with a median age of 84 years, IQR [77.0 to 91.0]. Women had a median age of 88 years compared to 81 years for men (*p* < 0.0001). The age distributions by sex are shown in the boxplot in Figure 1. Table 1 provides the anthropometric and sensory characteristics (hearing, vision).

When considering sex differences (Table 2), women had a higher degree of dependency on the Barthel Scale, ranging from mild to severe levels of dependency. This scale is an instrument used to assess independence in performing activities of daily living.

The same trend was observed with the Pfeiffer Scale, which assesses cognitive impairment, showing higher percentages of mild, moderate, and severe cognitive impairment in women compared to men.

The instrumental device used to assess frailty showed a higher percentage of disability in women than in men, with statistically significant differences.

Regarding fall risk assessment using the scales (Table 3), it was observed that the greater the degree of dependence on the Barthel Scale, the higher the fall risk.

The Charlson Comorbidity Index showed that the greater the comorbidity, the higher the fall risk, with a lower risk when there were no additional comorbidities.

The MNA nutritional questionnaire highlighted the importance of adequate nutritional status, as a better nutritional state reduced the fall risk. Thus, malnutrition significantly increased the percentage of falls.

The instrumental device assessing fall risk found a higher percentage of fall risk in the group with a greater risk of falling and a lower percentage in those with no fall risk.

Logistic regression models were established to relate how different questionnaires behaved in relation to fall risk (Table 4), finding that the Barthel Scale, considering its numerical scores, acted as a protective factor against fall risk with a significant result. A higher Barthel score indicates less dependency.

Additionally, the numerical scale of the MNA nutritional status questionnaire acted as a protective factor, with higher scores correlating to lower fall risk. Malnutrition states had lower scores.

The instrumental device that assessed fall risk considered a fall risk factor when numerical measurements were higher. The higher the score, the greater the fall risk detected.

## 4. Discussion

The findings of this study provide relevant evidence on the relationship between fall risk and frailty in institutionalized elderly individuals, highlighting the role of functional dependency, cognitive impairment, comorbidity, and nutritional status as determining factors.

The results obtained through the Barthel Scale indicate that the greater the functional dependency, the higher the fall risk, which aligns with previous studies by Nordling et al. [22]. Dependence on activities of daily living limits mobility and postural control, increasing the likelihood of falls. In this study, greater dependence was observed in women, which could explain the higher percentage of falls recorded in this population.

Cognitive impairment was also identified as a relevant factor, with a higher prevalence in women according to the Pfeiffer Scale. Unlike the findings of Yuan et al. [23], who established a strong association between physical frailty and cognitive impairment, this study found that cognitive impairment was more directly related to fall risk. These results suggest that interventions aimed at preserving cognitive function could indirectly contribute to fall prevention in institutionalized elderly individuals.

The analysis of comorbidity using the Charlson Scale showed that a higher number of comorbidities was associated with an increased risk of falls, in line with previous studies by Xie et al. [24]. This underscores the importance of comprehensive medical care to minimize the impact of chronic diseases on mobility and postural stability.

Nutritional status, assessed using the Mini Nutritional Assessment (MNA), acted as a protective factor, showing that better nutrition significantly reduces the risk of falls. These results are consistent with those of Adly et al. [25], who also found a significant relationship between malnutrition and falls. Adequate nutrition not only influences muscle mass and strength but also cognitive and functional capacity, so strategies focused on improving nutrition could have a positive impact on fall prevention [10].

The use of an instrumental device to measure frailty and fall risk allowed for an objective assessment, revealing significant differences between men and women. This device demonstrated its ability to identify participants with higher vulnerability, highlighting its potential utility in clinical and geriatric settings. Previous studies, such as that of Calderón Rojas et al. [26], have employed similar technology, although focusing exclusively on frailty. The combination of both assessments in a single device represents a key methodological innovation. Like our device, Amacho et al. [27] emphasized the importance of assessing frailty in primary care using specific indices, such as the frailty-VIG index, which has shown convergent and discriminative validity compared to the Short Physical Performance Battery (SPPB) in the general population over 70 years of age. Likewise, the study by Vera-Remartínez et al. [15] demonstrated the validity and reliability of an Android-based device for assessing fall risk in elderly prisoners, highlighting the potential of technology for the early detection of fall risk in various settings. Their findings reinforce the need to use digital tools that enable accessible and accurate assessments, facilitating the implementation of preventive measures in vulnerable populations.

The results reinforce the need to implement multifactorial interventions that include physical exercise programs, nutritional strategies, and comorbidity control to reduce fall risk in older adults. Recent studies by Guirguis-Blake et al. [28] and Orts Cortes et al. [29] have demonstrated the effectiveness of these interventions, with significant reductions in fall incidence. Moreover, the study by Llaneras Gelabert [30] highlights the benefits of physical exercise using video games in fall prevention programs for elderly individuals. This innovative strategy not only improves adherence to physical activity but also enhances balance, strength, and coordination—key factors in reducing fall risk. The incorporation of interactive technologies into prevention programs could represent an effective and motivating alternative for promoting physical activity in this population, thereby maximizing the positive effects of multifactorial interventions.

Additionally, the importance of developing standardized care protocols and using technology for the early detection of frailty and fall risk is highlighted. General preventive measures, such as promoting regular physical exercise, balanced nutrition, reviewing polypharmacy, and adapting the environment, should be priorities in the care of institutionalized elderly individuals [31].

Among the main limitations of the study, the lack of validated tools for the simultaneous detection of frailty and fall risk stands out. One of the main limitations lies in a certain gender imbalance within the sample, with a higher proportion of women. Additionally, the average age among female participants is also higher than that of their male counterparts.

Nonetheless, the main strength of this study lies in the development and application of an innovative device capable of assessing both factors in an integrated manner, facilitating the identification of at-risk patients and allowing personalized interventions to improve their quality of life.

## 5. Conclusions

The development of technological applications like Oldfry, which assess frailty and fall risk in older adults, is crucial for preventing these events and their adverse consequences. Moreover, it allows healthcare professionals to identify specific areas of weakness. This proactive approach can significantly improve the quality of life for older adults by reducing the risk of falls and promoting healthier and more autonomous aging.

The Oldfry device assessed fall risk at a higher percentage in the group presenting a higher fall risk, in addition to considering fall risk as a risk factor when higher numerical measurements were obtained. Similarly, the Barthel Scale and the MNA Scale were observed as protective factors for fall risk in a significant way. Therefore, it is essential to consider all the related factors, as the analysis shows that functional status, comorbidities, and nutritional status are directly related to falls and frailty.

Overall, it is planned to implement physical activity sessions in residential care centers to promote physical health and autonomy among older adults, in addition to tracking the Oldfry tool to monitor participant outcomes over time.

## Figures and Tables

**Figure 1 sensors-25-02964-f001:**
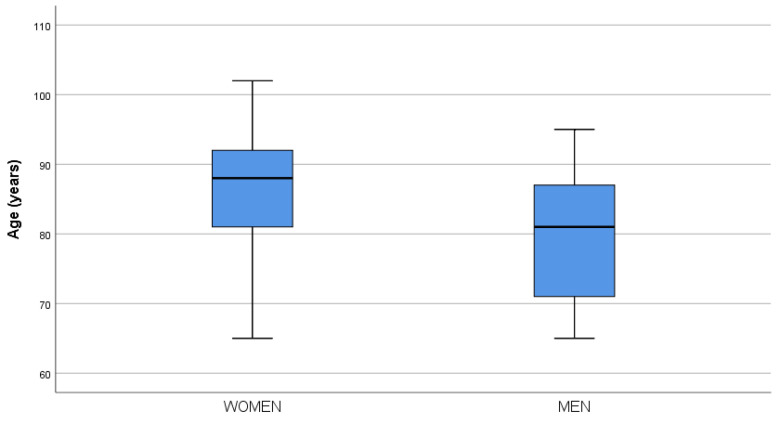
Boxplot age by sex.

**Table 1 sensors-25-02964-t001:** Descriptive analysis of the sample.

**VARIABLES:**	**CATEGORIES:**	**N (%)**	
Sex:	Men	62 (39.74%)	
	Women	94 (60.26%)	
Reported falls:	Yes	76 (48.72%)	
	No	80 (51.28%)	
Vision problems:	Yes	62 (39.74%)	
	No	94 (60.26%)	
Hearing problems:	Yes	33 (21.15%)	
	No	123 (78.85%)	
**VARIABLES:**	**CATEGORIES:**	**N (%)**	**Mean (IC: 95%) OR Median [P25–P75] ***
Weight (kg)	Underweight	2 (1.28%)	64.8 [56.78–76.00] *
	Normal weight	58 (37.18%)	
	Overweight	58 (37.18%)	
	Obesity	38 (24.36%)	
BMI (kg/m^2^)			26.77 (26.03–27.51)
Height (cm)			158.17 (156.96–159.39)
Age (years)			84 [77.0–91.0] *

CI: confidence interval; (*): corresponds to the expression of medians and interquartile range; P25: 25th Percentile; P75: 75th Percentile; BMI: Body Mass Index.

**Table 2 sensors-25-02964-t002:** Comparison of scales, fall risk, and frailty by sex.

SCALES:		TOTAL *n* = 156 (100%)	WOMEN *n* = 94 (60.26%)	MEN *n* = 62 (39.74%)	SIGNIFICANCE X^2^
YESSAVAGE	Normal	132 (84.62%)	82 (52.57%)	50 (32.05%)	0.264
	Moderate depression	19 (12.18%)	10 (6.41%)	9 (5.77%)	0.469
	Severe depression	5 (3.21%)	2 (1.28%)	3 (1.93%)	0.347
BARTHEL	Independent	21 (13.46%)	4 (2.56%)	17(10.90%)	<0.0001
	Mild dependence	114 (73.07%)	76 (48.71%)	38 (24.36%)	0.007
	Moderate dependence	19 (12.18%)	12 (7.69%)	7 (4.49%)	0.781
	Severe dependence	2 (1.28%)	2 (1.28%)	0 (0.0%)	0.248
PFEIFFER	Normal	51 (32.69%)	23 (14.74%)	28 (17.95%)	0.007
	Mild deterioration	39(25.00%)	34(21.79%)	5(3.21%)	<0.0001
	Moderate deterioration	54 (34.62%)	32 (20.52%)	22 (14.10%)	0.853
	Severe deterioration	12 (7.69%)	5(3.20%)	7 (4.49%)	0.171
CHARLSON	High	72 (46.15%)	40 (25.64%)	32 (20.51%)	0.266
	Low	25 (16.03%)	19 (12.,18%)	6 (3.85%)	0.079
	None	59 (37.82%)	35 (18.38%)	24 (12.61%)	0.852
MNA TOTAL	Normal	90 (57.69%)	56 (35.90%)	34 (21.79%)	0.001
	Risk of malnutrition	56 (35.90%)	23 (14.74%)	33 (21.15%)	0.056
	Malnutrition	10 (6.41%)	1 (0.64%)	9 (5.77%)	0.007
MEDICATIONS	Low consumption	21 (13.46%)	10 (6.41%)	11 (7.05%)	0.718
	Medium consumption	8 (5.13%)	6 (3.85%)	2 (1.28%)	0.168
	High consumption	127 (81.41%)	64 (41.03%)	63 (40.38%)	0.642
FALL RISK (Oldfry device)	Normal	100 (64.10%)	63 (40.38%)	37 (23.72%)	0.349
	Mild	25 (16.03%)	11 (7.05%)	14 (8.98%)	0.070
	High	31 (19.87%)	20 (12.82%)	11 (7.05%)	0.588
FRAILTY (Oldfry device)	Independent	25 (16.03%)	15 (9.62%)	10 (6.41%)	0.977
	Pre-fragile	17(10.90%)	10 (6.41%)	7 (4.49%)	0.898
	Fragile	79 (50.64%)	42 (26.92%)	37 (23.72%)	0.067
	Disabled	35 (22.43%)	27 (17.30%)	8 (5.13%)	0.020

X^2^: Chi-square. MNA: Mini Nutritional Assessment.

**Table 3 sensors-25-02964-t003:** Comparison of scales, fall risk, and frailty in relation to falls experienced.

SCALES:		TOTAL *n* = 156 (100%)	NO FALLS *n* = 80 (51.28%)	FALLS *n* = 76 (48.72%)	SIGNIFICANCE X^2^
YESSAVAGE	Normal	132 (84.62%)	68 (43.59%)	64 (41.03%)	0.321
	Moderate depression	19 (12.18%)	11 (7.05%)	8 (5.13%)	0.662
	Severe depression	5 (3.21%)	4 (2.57%)	1 (0.64%)	0.222
BARTHEL	Independent	21 (13.46%)	18 (11.54%)	3 (1.92%)	0.0007
	Mild dependence	114 (73.07%)	53 (33.97%)	61 (39.10%)	0.049
	Moderate dependence	19 (12.18%)	8 (5.13%)	11(7.05%)	0.393
	Severe dependence	2 (1.28%)	1 (0.64%)	1 (0.64%)	0.971
PFEIFFER	Normal	51 (32.69%)	30 (19.23%)	21 (13.46%)	0.189
	Mild deterioration	39(25.00%)	18 (11.54%)	21 (13.46%)	0.459
	Moderate deterioration	54 (34.62%)	23 (16.67%)	31 (19.87%)	0.114
	Severe deterioration	12 (7.69%)	9(5.77%)	3(1.92%)	0.087
CHARLSON	High	72 (46.15%)	30 (19.23%)	42 (26.92%)	0.026
	Low	25 (16.03%)	13 (8.34%)	12 (7.69(%)	0.937
	None	59 (37.82%)	37 (23.72%)	22 (14.10%)	0.005
MNA TOTAL	Normal	90 (57.69%)	56 (35.90%)	34 (21.79%)	0.558
	Risk of malnutrition	56 (35.90%)	33 (21.15%)	23 (14.74%)	0.799
	Malnutrition	10 (6.41%)	5 (3.21%)	5 (3.21%)	0.493
MEDICATIONS	Low consumption	21 (13.46%)	12 (7.69%)	9 (5.77%)	0.609
	Medium consumption	8 (5.13%)	5 (3.21%)	3 (1,92%)	0.984
	High consumption	127 (81.41%)	80 (51.28%)	47 (30.13%)	0.661
FALL RISK (Oldfry device)	Normal	100(64.10%)	58 (37.18%)	42 (26.92%)	0.025
	Mild	25 (16.03%)	14 (8.98%)	11 (7.05%)	0.607
	High	31 (19.87%)	8 (5.13%)	23 (14.74%)	0.002
FRAILTY (Oldfry device)	Independent	25 (16.03%)	16 (10.26%)	9 (5.77%)	0.165
	Pre-fragile	17(10.90%)	11 (7.05%)	6 (3.85%)	0.241
	Fragile	79 (50.64%)	39 (25.00%)	40 (25.64%)	0.628
	Disabled	35 (22.43%)	14 (8.97%)	21 (13.46%)	0.129

X^2^: Chi-square. MNA: Mini Nutritional Assessment.

**Table 4 sensors-25-02964-t004:** Logistic regression model for fall risk and frailty.

VARIABLES:	O.R.	IC 95%	SIGNIFICANCE
Yessavage (Numeric Scale)	0.887	−0.238 to 0.059	0.249
Barthel (Numeric Scale)	0.983	−0.047 to −0.002	0.035
Pfeiffer (Numeric Scale)	1.021	−0.153 to 0.109	0.751
MNA (Numeric Scale)	0.794	−0.339 to −0.101	0.0004
Charlson (Numeric Scale)	1.099	−0.095 to 0.257	0.375
Number of medications	1.065	−0.0001 to 0.177	0.054
Fall risk (Oldfry device)	1.172	0.054 to 0.248	0.004
Fragility risk (Oldfry device)	1.111	−0.188 to 0.411	0.487
Body Mass Index (BMI)	0.963	−0.119 to 0.055	0.369
Age	1.023	0.987 to 1.059	0.214

O.R.: Odds Ratio; 95% CI: 95% confidence interval.

## Data Availability

The data supporting the findings of this study are available upon reasonable request from the corresponding author. Due to privacy and ethical restrictions, the data are not publicly accessible.

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
