# Peer review of "Detection of Falls and Frailty in Older Adults with Oldfry: Associated Risk Factors"

_sensors, 2025, doi:10.3390/s25102964_

Round 1
Reviewer 1 Report
Comments and Suggestions for Authors
This manuscript states that Oldfry technology is an innovative digital tool developed to assess and monitor frailty and fall risk in older people, and it explains the results. However, as the authors mentioned in Chapter 2, the validation of the Oldfry tool has been completed, and the paper is under publication. As one of the reviewers, I believe it isn't easy to evaluate the technical usefulness of Oldfry technology based only on the results presented by the authors without a detailed understanding of the technology/tool. Therefore, it would be appropriate to re-review this manuscript later based on the publication of the document mentioned in Chapter 2. If not, and if the manuscript is to be evaluated solely, I think it would be appropriate to include a detailed introduction of the Oldfry technology/tool ​​in the introduction.
Author Response
→ A detailed introduction to the functioning of the Oldfry tool has been included in the manuscript’s Introduction. Furthermore, validation data has been added in the Methodology section.
Reviewer 2 Report
Comments and Suggestions for Authors
Strengths:
1. Strong relevance of research topic: With the aging population, frailty and fall prevention are important issues in current geriatric medicine, giving the research significant clinical value.
2. Sound methodological design: The selection of appropriate sample size, age composition, and multiple standardized assessment tools enhances the reliability of the results.
3. Innovation: The Oldfry device's method of simultaneously assessing frailty and fall risk is innovative, providing new approaches for health monitoring in older adults.
4. Comprehensive data analysis: The research employs appropriate statistical methods, including descriptive analysis, bivariate analysis, non-parametric tests, and binary logistic regression model analysis, to comprehensively evaluate various factors.
5. Clear results: The study clearly identifies functional dependency, nutritional status, and comorbidities as key factors influencing fall risk, providing clear guidance for clinical practice.
Areas for improvement:
1. Gender imbalance in the sample: Women constitute 60.26% of the sample, and their average age is significantly higher than men's, which may affect the generalizability of the research results. It is recommended to acknowledge this limitation in the discussion section and consider conducting gender-specific analyses or adjusting for this confounding factor.
2. Insufficient description of the Oldfry device: The paper lacks detailed description of the technical specifications, operational methods, and validation process of the Oldfry device, making it difficult to assess its reliability and effectiveness. It is suggested to add relevant content or cite previous validation studies.
Format issues:
1. Inconsistent table formatting: Tables 1, 2, and 3 are inconsistently formatted, with chaotic data alignment. It is recommended to standardize table formats and ensure data alignment.
2. Inconsistent bullet point formatting in line 119.
3. Inconsistent reference formatting in lines 342 and 409.
4. Figure 1 caption placement: Should the caption be positioned below the figure?
Specific recommendations:
1. The introduction should more clearly articulate the uniqueness of the Oldfry device and its differences from existing assessment tools.
2. The methods section should provide detailed explanation of the Oldfry device's technical principles, measurement indicators, and validation process.
3. The results section should consider adding data analyses adjusted for gender and age to reduce the influence of confounding factors.
4. The discussion section should include more on the limitations of sample representativeness and the applicability of results across different gender and age groups.
Overall evaluation:
This research explores an important topic in geriatric medicine and provides valuable data supporting the application of technology-assisted assessment in preventing falls among older adults. Despite some methodological limitations and formatting issues, the research results still have certain clinical guiding significance. It is recommended that the authors resubmit after making appropriate modifications based on the above comments.
Author Response
Areas for improvement:
- Gender imbalance in the sample: Women make up 60.26% of the sample, and their average age is significantly higher than that of men, which may affect the generalizability of the research findings. It is recommended to acknowledge this limitation in the Discussion section and consider conducting gender-specific analyses or adjusting for this confounding factor.
→ As suggested by the reviewer, the gender imbalance in the sample, as well as
the average age difference for women, has been included in the Discussion
section under the limitations.
- Insufficient description of the Oldfry device: The article lacks a detailed description of the technical specifications, operating methods, and validation process of the Oldfry device, making it difficult to evaluate its reliability and effectiveness. It is suggested to add relevant content or cite previous validation studies.
→ Technical specifications and operating methods of the Oldfry device are explained in
the Introduction. In the Methodology section, under instruments, validation data from
the prior study (pending publication) has been provided.
Formatting Issues:
- Inconsistent table formatting: Tables 1, 2, and 3 show inconsistent formatting, with chaotic data alignment. It is recommended to standardize table formats and ensure proper alignment of the data.
→ The tables have been reviewed and replaced. Consistent formatting has
been applied.
- Inconsistent bullet formatting in line 119.
→ Corrected and formatted uniformly using bullet dashes. - Inconsistent reference formatting in lines 342 and 409.
→ Corrected. - Placement of Figure 1 legend: Should the caption be placed below the figure?
→ The figure legend has been placed as a caption below the figure.
Specific Recommendations:
- The Introduction should more clearly articulate the uniqueness of the Oldfry device and how it differs from existing assessment tools.
→ This has been done following the criteria of both reviewers. - The Methods section should provide a detailed explanation of the device’s technical principles, measurement indicators, and validation process.
→ Done. - The Results section should consider including data analysis adjusted by sex and age to reduce the influence of confounding factors.
→ Not addressed. A sex-based comparison is already included in the bivariate analysis. - The Discussion section should include more information about the limitations in sample representativeness and the applicability of results to different sex and age groups.
→ Addressed in the Limitations section.
Overall Evaluation:
This research explores an important topic in geriatric medicine and provides valuable data supporting the use of technology-assisted assessment in fall prevention among older adults. Despite some methodological limitations and formatting issues, the research findings still hold clinical relevance. It is recommended that the authors resubmit the manuscript after making the necessary modifications based on the above comments.